# Defects in lipid homeostasis reflect the function of TANGO2 in phospholipid and neutral lipid metabolism

Agustin Leonardo Lujan[1], Ombretta Foresti[1], Conor Sugden[1], Nathalie Brouwers[1], Alex Mateo Farre[1], Alessio Vignoli[1], Mahshid Azamian[2], Alicia Turner[3], Jose Wojnacki[1], Vivek Malhotra[1,4,5]*

[1]Centre for Genomic Regulation (CRG), The Barcelona Institute for Science and Technology, Barcelona, Spain; [2]Center for Cell and Gene Therapy, Baylor College of Medicine, Houston, United States; [3]Department of Molecular and Human Genetics, Baylor College of Medicine, Houston, United States; [4]Universitat Pompeu Fabra (UPF), Barcelona, Spain; [5]ICREA, Barcelona, Spain

**Abstract** We show that TANGO2 in mammalian cells localizes predominantly to mitochondria and partially at mitochondria sites juxtaposed to lipid droplets (LDs) and the endoplasmic reticulum. HepG2 cells and fibroblasts of patients lacking TANGO2 exhibit enlarged LDs. Quantitative lipidomics revealed a marked increase in lysophosphatidic acid (LPA) and a concomitant decrease in its biosynthetic precursor phosphatidic acid (PA). These changes were exacerbated in nutrient-starved cells. Based on our data, we suggest that TANGO2 function is linked to acyl-CoA metabolism, which is necessary for the acylation of LPA to generate PA. The defect in acyl-CoA availability impacts the metabolism of many other fatty acids, generates high levels of reactive oxygen species, and promotes lipid peroxidation. We suggest that the increased size of LDs is a combination of enrichment in peroxidized lipids and a defect in their catabolism. Our findings help explain the physiological consequence of mutations in TANGO2 that induce acute metabolic crises, including rhabdomyolysis, cardiomyopathy, and cardiac arrhythmias, often leading to fatality upon starvation and stress.

*For correspondence:
vivek.malhotra@crg.eu

## Editor's evaluation

This important manuscript describes a series of cellular phenotypes associated with the depletion of TANGO2, a poorly characterized gene product but relevant to neurological and muscular disorders. The authors present solid data indicating that TANGO2 associates with membrane-bound organelles, mainly mitochondria, impacting lipid metabolism and the accumulation of reactive oxygen species.

## Introduction

In 2006, we reported a collection of new genes required for transport and organization of the Golgi complex in *Drosophila* (*Bard et al., 2006*). These genes, monikered TANGO, include TANGO1, a transmembrane protein required for the organization of endoplasmic reticulum (ER) exit sites where it collects bulky molecules like collagens and stabilizes transient inter-organelle tunnels for their export to the next compartment of the secretory pathway (*Raote and Malhotra, 2021*).

TANGO2, a gene unrelated to TANGO1, lacks a transmembrane domain and is reported to be cytosolic and also localized to mitochondria (*Jennions et al., 2019*; *Milev et al., 2021*). TANGO2

affects ER-to-Golgi transport and mitochondria physiology; however, its precise role in these compartments is unknown (*Mingirulli et al., 2020*; *Bérat et al., 2021*; *Milev et al., 2021*). Mutations in TANGO2 result in metabolic encephalopathy and arrhythmias and these conditions are exacerbated upon nutrient starvation, often leading to fatality (*Kremer et al., 2016*; *Lalani et al., 2016*). What is the physiological role of TANGO2, and how do mutations in TANGO2 lead to fatality in conditions of starvation?

We show here that TANGO2 in mammalian cells localizes predominantly to mitochondria, but also partially to sites where the ER and lipid droplets (LDs) are juxtaposed to mitochondria. Knockdown of TANGO2 increased the size of LDs and elevated intracellular reactive oxygen species (ROS) levels. Mass spectrometric quantification of cellular lipids revealed that TANGO2-deficient cells exhibited markedly high lysophosphatidic acid (LPA) and low levels of phosphatidic acid (PA). These cells were also highly reduced in cardiolipin (CL), which is ordinarily produced from PA. Moreover, changes in these cellular properties were further exacerbated in cells cultured in low-nutrient media. Many of these features match properties in cells derived from patients with mutations in TANGO2. Based on our data, we propose that TANGO2 functions in lipid homeostasis at the level of acyl-CoA metabolism, and defects in lipid metabolism are the main cause of starvation-induced acute rhabdomyolysis, cardiomyopathy, and cardiac arrhythmias.

## Results

### TANGO2 localizes predominantly to the mitochondria and transiently to LDs and ER

In humans, six isoforms of TANGO2 are produced by alternative splicing. The TANGO2-1 (TNG2_HUMAN·Q6ICL3-1) and TANGO2-2 (TNG2_HUMAN·Q6ICL3-2) isoforms are most similar to their orthologs, such as hrg-9 in worms, ygr127w in yeast, and tango2 in zebrafish (*Figure 1A, B*). It is important to note that the anti-Tango2 antibody we have used here recognizes both isoforms in human cells, and tagged versions of both isoforms show the same intracellular location (*Figure 1A*). As shown in the following sections, siRNA-based depletion affects all isoforms. Therefore, the effect of TANGO2 depletion on cell physiology is a result of reduction in the levels of all isoforms.

Previous published data showed conflicting results on the intracellular location of TANGO2 (*Jennions et al., 2019*; *Heiman et al., 2022*; *Milev et al., 2021*). Unfortunately, the commercially anti-TANGO2 antibodies are not suitable for immunofluorescence analysis. Therefore, we monitored the location of TANGO2 distribution dynamically in living cells. We generated C-terminally tagged version of Tango2 by in frame fusion with either mScarlet (TANGO2.Iso1-mScarlet) or EGFP (TANGO2.Iso1-EGFP). By transient transfections in HepG2 cells, we co-expressed these forms of TANGO2 with fluorescent markers of various subcellular organelles including, mitochondria (Mito-mTurquoise), ER (ER-mTurquoise), peroxisomes (Peroxisome-mTurquoise), and LDs (GPAT4-mNeonGreen). All images were acquired by live-cell confocal time-lapse microscopy to avoid fixation artefacts. We observed high colocalization (Pearson coefficient, $r = 0.88$) between TANGO2 and the mitochondria marker. TANGO2 was also enriched at sites of mitochondria closely juxtaposed to ER and LDs (*Figure 2A*). To further ascertain the location of TANGO2, we monitoried its intracellular locale with respect to ER and LDs in fixed cells. HepG2 cells were co-transfected with TANGO2-mScarlet and VAPB-NeonGreen – a protein located in the ER membrane and in the mitochondria-associated membrane sites (*Figure 2—figure supplement 1A*), or with the LD marker GPAT4-NeonGreen (*Figure 2—figure supplement 1B*). We observed TANGO2-Scarlet and the specific markers of ER and LDs in close position in *XY*, *XZ*, and *YZ* planes.

To confirm the precise topology of TANGO2-mScarlet in the mitochondria membrane, we evaluated the proximity between TANGO2 and Tom20 – a protein marker of the outer mitochondrial membrane (OMM) – by Förster resonance energy transfer (FRET) microscopy in fixed cells. This procedure analyzes the radiation energy transfer from an excited fluorescent molecule (called the donor) to an acceptor molecule; this occurs when both molecules are within a range of 1–10 nm from each other (*Broussard et al., 2013*). To explore the proximity of TANGO2 toTom20, HepG2 cells were transfected with TANGO2.mScarlet (red), fixed, and stained with anti-Tom20 antibody (green) followed by Alexa 633 secondary antibody. After photobleaching (white square) of Alexa 633 we observed a

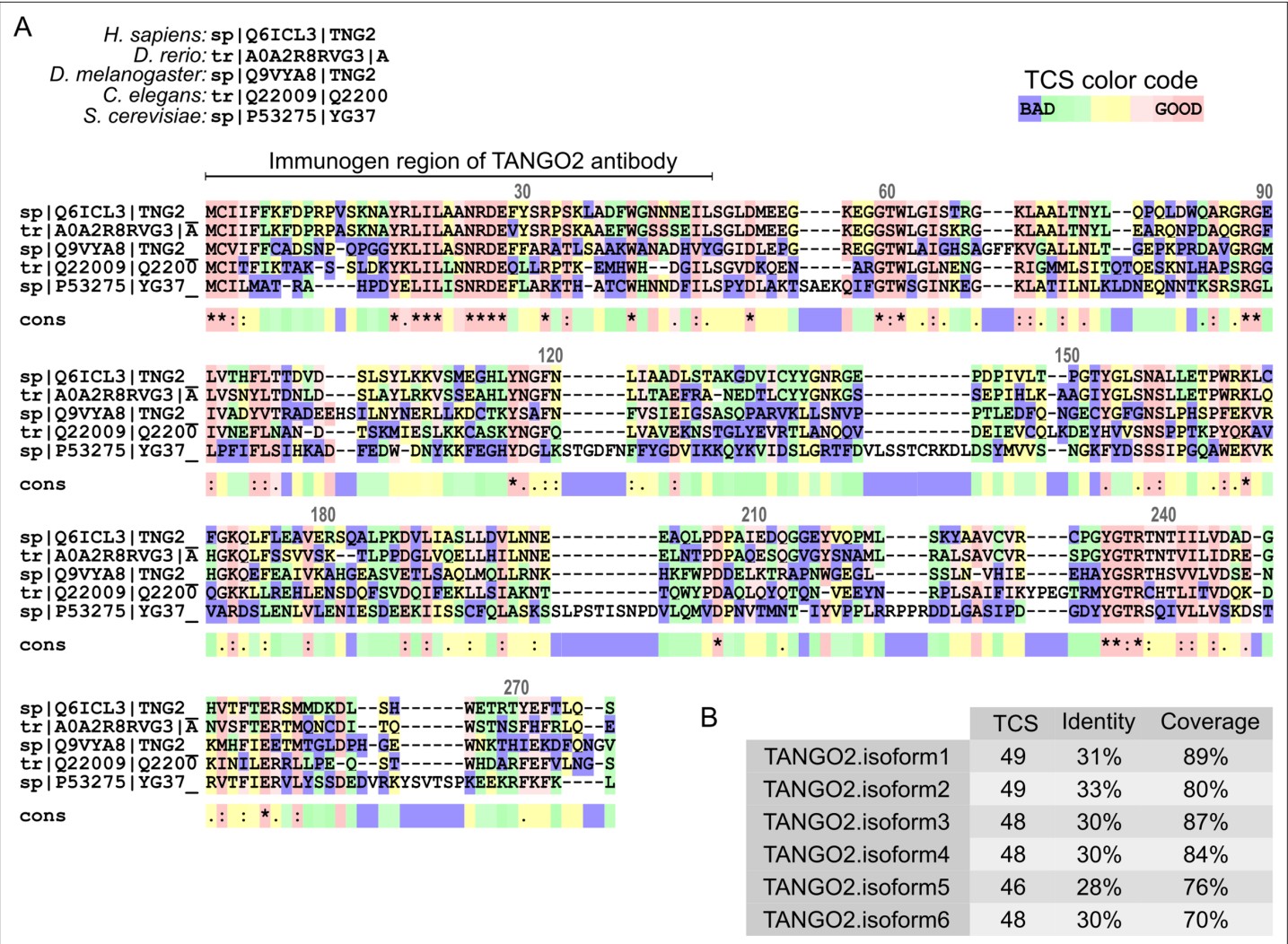

**Figure 1.** TANGO2 orthologs. (**A**) In silico sequence alignment of TANGO2 orthologs in human (*H. sapiens*), zebrafish (*D. rerio*), fruit fly (*D. melanogaster*), worm (*C. elegans*), and yeast (*S. cerevisiae*) using T-COFFEE software. The transitive consistency score (TCS) color code indicates the agreement between the library and the considered alignment. The TCS scale goes from blue (less consistent) to dark pink (high reliability and more accurate agreement) brick regions. The color line (cons?) below the alignment indicates the consensus score of every column. Residue numbering is based on the full length of the human TANGO2 protein (TNG2_HUMAN·Q6ICL3). (**B**) Multiple sequence alignment (MSA) of each human TANGO2 isoform with species orthologs. The TCS identifies the most correct alignment positions in MSA, the Identity evaluates the sequence function, and the Coverage describes the average number of reads that align to known reference bases.

significant increase (p=0.0002) in mScarlet-fluorescence intensity compared to an unbleached region (yellow square), confirming an FRET signal (***Figure 2B–D***).

To avoid potential artefacts from an uneven or abnormal protein expression by transient transfection, we generated stable cell lines expressing low levels of the C-terminally EGFP-tagged TANGO2 using lentiviral infection (***Tandon et al., 2018***). Live imaging of stable cell lines revealed that TANGO2-GFP localizes predominantly to mitochondria (***Figure 2—figure supplement 1C***). We also observed a diffuse cytosolic staining (***Figure 2—figure supplement 1D***). To check whether TANGO2-EGFP is cleaved in our experimental conditions giving rise to different forms, we verified the size of expressed construct by sodium dodecyl sulfate–polyacrylamide gel electrophoresis (SDS–PAGE) followed by western blotting with anti-GFP antibody (***Figure 2—figure supplement 1E***). This revealed a single band that matches the expected molecular weight of EGFP-tagged TANGO2.

It is therefore reasonable to conclude that despite lacking a transmembrane domain, TANGO2 is attached predominantly to the mitochondrial outer membrane and at sites of mitochondria that are

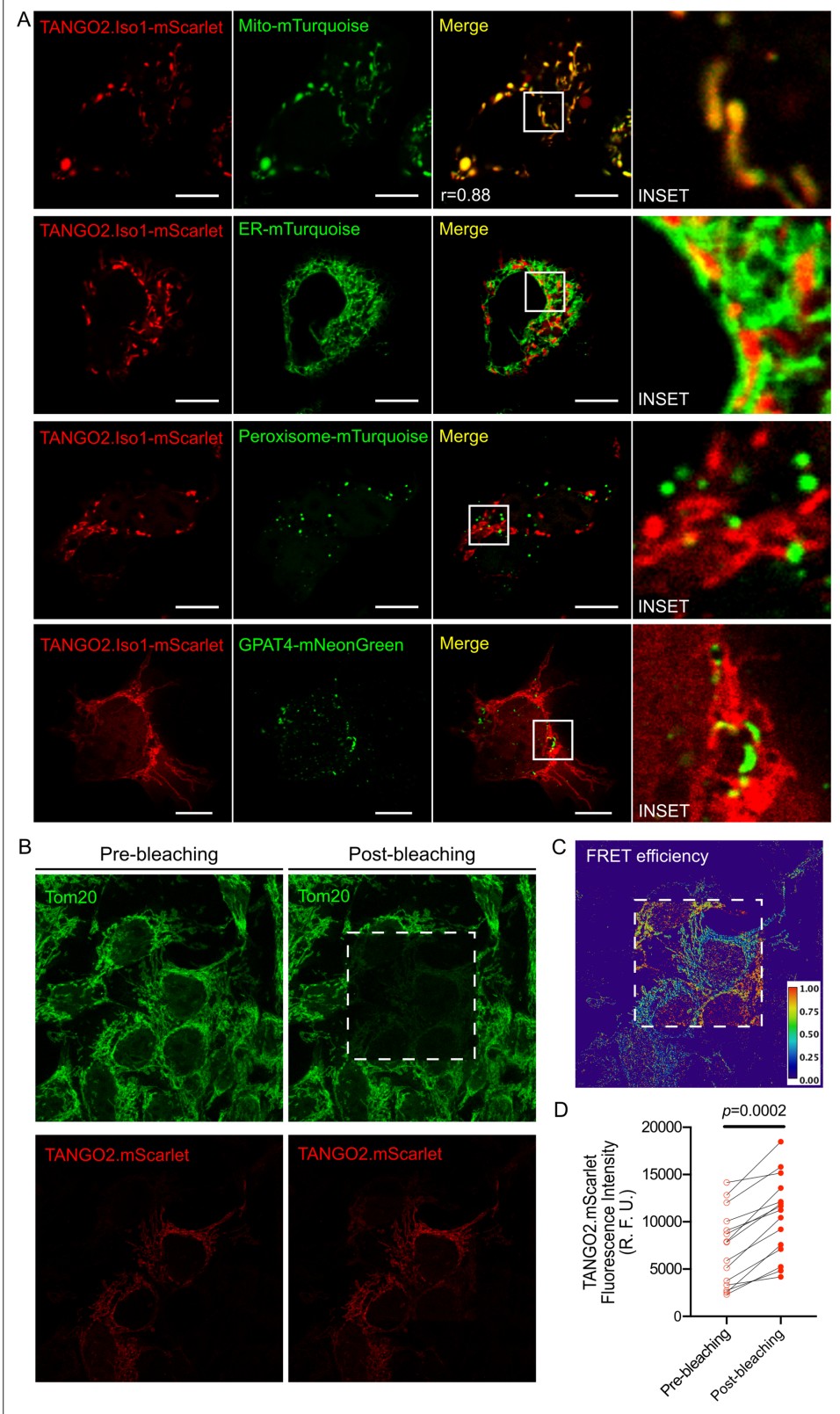

**Figure 2.** TANGO2 location in HepG2 cells. (**A**) Cells expressing TANGO2.Iso1-mScarlet co-transfected with Mitochondria-pmTurquoise2 (Mito-mTurquoise), ER-pmTurquoise2 (ER-mTurquoise), Peroxisome-SKL-mTurquoise2 (Peroxisome-mTurquoise), or GPAT4-hairpin-mNeonGreen (GPAT4-mNeonGreen) to detect lipid droplets. The yellow color indicates colocalization between both red and green labels. Pearson coefficient (*r*) was calculated with

*Figure 2 continued on next page*

*Figure 2 continued*

the coloc2 plugin in ImageJ software. Images are representative of three independent experiments. Scale bars = 10 μm. (**B–D**) HeLa cells were transfected with TANGO2.mScarlet, fixed, and incubated with anti-Tom20 followed by Alexa 633 secondary antibodies. (**B**) Representative image of the fluorescence intensity in the acceptor (upper panels) and donor (bottom panels) channels before (left panels) and after photobleaching (right panels) of the acceptor. (**C**) Representative map of Förster resonance energy transfer (FRET) efficiency calculated from the donor fluorescence intensity as described in Materials and methods. (**D**) Quantification of the increase in fluorescence intensity of the donor channel (TANGO2.mScarlet) after photobleaching (*n* = 15 cells). R. F. U. means relative fluorescence units.

The online version of this article includes the following figure supplement(s) for figure 2:

**Figure supplement 1.** Intact expressed tagged TANGO2 is in close proximity to endoplasmic reticulum (ER) and lipid droplets (LDs).

juxtaposed to the ER and the LDs. TANGO2 likely cycles between the cytoplasm and the membrane. The mechanism of how TANGO2 is recruited to the membrane is not known.

## TANGO2-lacking cells contain larger LDs

As shown above, TANGO2 frequently appears at sites that are marked by LDs. This prompted us to test whether cells lacking TANGO2 are affected in any aspect of LDs. We silenced TANGO2 (siTANGO2) by standard siRNA-based procedures. A control was generated with non-targeting siRNA and described henceforth as Mock. The silencing procedure reduced TANGO2 levels by ~45% after one round of siRNA transfection. However, after two sequential rounds of transfection, a ~75% reduction in TANGO2 levels was evident (*Figure 3—figure supplement 1A*). We have used double sequential transfections to reduce TANGO2 levels in all experiments that follow (*Figure 3—figure supplement 1B*). Mock HepG2 cells and TANGO2-depleted cells were cultured in medium (Dulbecco's Modified Eagle's Medium, DMEM) supplemented with 10% fetal bovine serum (FBS) (DMEM complete) or OptiMEM without FBS (starvation) for 4 hr. Cells were then incubated with membrane permeant neutral lipid marker HCS LipidTOX Deep Red (red) and the DNA marker Hoechst-33342 (blue) for 30 min at 37°C followed by live-cell imaging. LDs were consistently larger in TANGO2-depleted cells compared to mock cells (*Figure 3A*, left panel). Moreover, nutrient-deprived TANGO2-depleted cells (Starvation; *Figure 3A*, right panel) had even larger LDs. To confirm that these structures are indeed LDs, TANGO2-depleted HepG2 cells were fixed and visualized with an antibody to the LD-resident protein ADRP (red). Nuclear DNA in these cells was labeled with 4',6-diamidino-2-phenylindole (DAPI; blue). There was indeed an increase in the size of ADRP-containing LDs upon depletion of TANGO2 (*Figure 3B*). We developed a macro for ImageJ Software, to quantify this increased retention of neutral lipids upon depletion of TANGO2. This procedure analyzes *z*-stack images to determine the number and volume of each LipidTOX-positive particle. Five samples were used for each condition, with a total of 67 Mock cells and 75 TANGO2-depleted cells, respectively. As shown in *Figure 3*, total volume of LDs per cell (panel C) and the volume of each LD (panel D) were significantly higher in TANGO2 depleted compared to the Mock cells (p = 0.0005 and p = 0.002, respectively). However, the number of LDs in each cell remains the same regardless of the TANGO2 levels (*Figure 3E*). Therefore, the increase in size is not caused by fusion of smaller LDs under these experimental conditions. To further analyze the effect of TANGO2 on LDs size, we added 120 nM Oleic acid (OA), which is known to potentiate LD formation, at 37°C for 6 hr to TANGO2 transiently depleted HepG2 cells (Mock vs. siTANGO2) and fibroblasts from TANGO2 deficiency disorder (TDD) donors (WT vs. TAN043). Changes in the total volume of LDs were evaluated by fluorescence intensity (FI) of allophycocyanin (APC) corresponding to HCS LipidTOX with flow cytometry (*Figure 3—figure supplement 1C*). Three independent experiments were analyzed for each condition in triplicate. The loss of TANGO2 revealed a significant increase in total LDs volume compared to Mock HepG2 cells (*Figure 3F*) or wild-type fibroblasts (*Figure 3G*). To reveal the cause for increased volume of LDs we analyzed their metabolism. To explore LD biogenesis in WT and TAN043 fibroblast, we generated three independent medium conditions of DMEM complete (control), adding 120 nM OA at 37°C for 6 hr in DMEM complete (+OA), or adding 120 nM OA at 37°C for 6 hr with the lipolysis inhibitor diethylumbelliferyl phosphate (DEUP) in DMEM complete (+OA + DEUP) (*Figure 3—figure supplement 1D*). The inhibition of lipolysis in

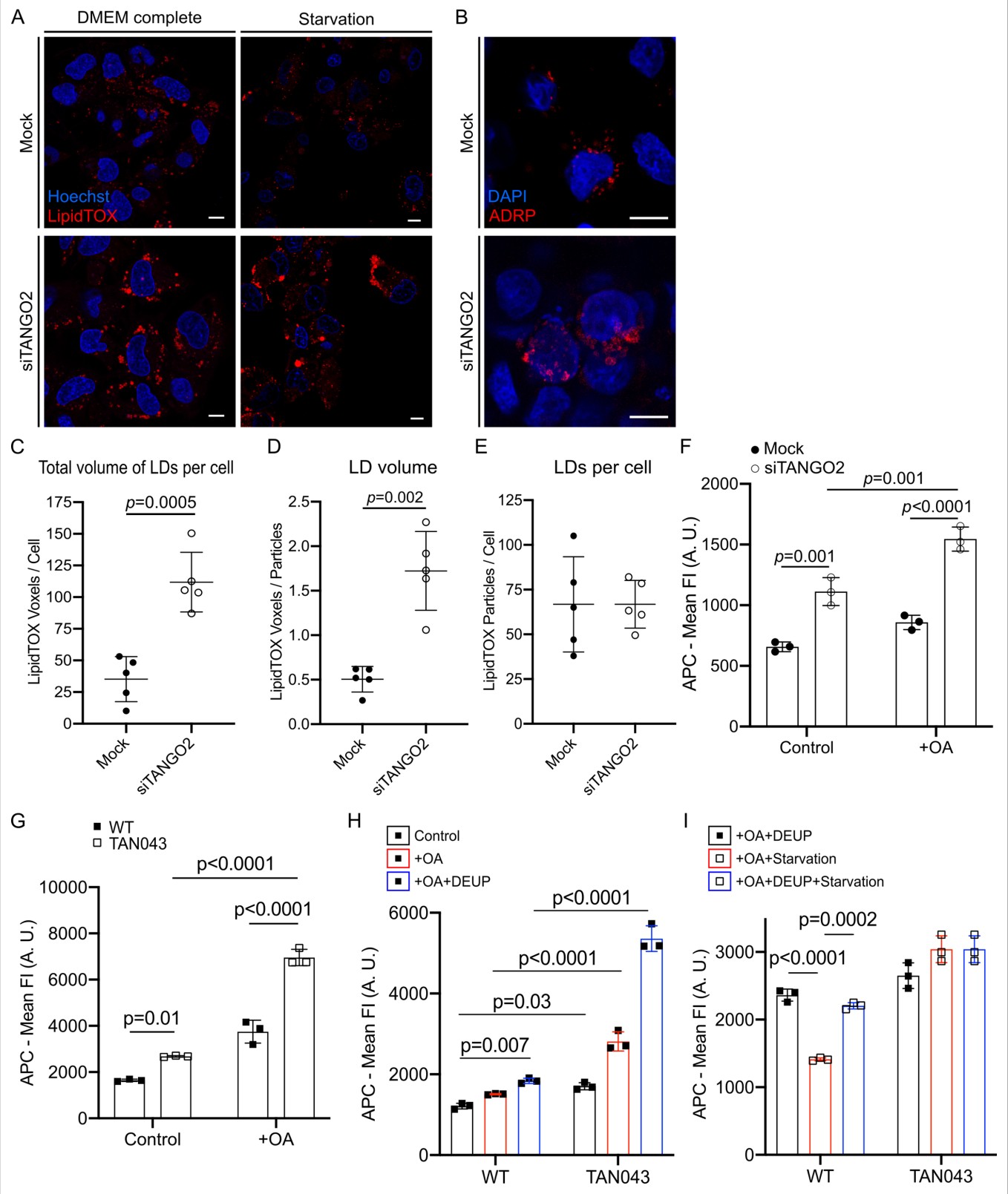

**Figure 3.** TANGO2 depletion affects lipid droplets' (LDs) size, biogenesis, and lipolysis. (**A**) Confocal images of HCS LipidTOX Deep Red marker (red) and Hoechst-33342 (blue) in mock (top) and TANGO2-depleted (bottom) HepG2 cells cultured in control conditions (Dulbecco's Modified Eagle's Medium [DMEM] complete; left panel) or starved of nutrients (starvation; right panel). Scale bars = 10 µm. (**B**) Mock (top) and TANGO2-depleted (bottom) cells in control conditions were fixed and incubated with anti-ADRP (red) and DAPI (blue). Scale bars = 10 µm. (**C–E**) Quantification of LDs

*Figure 3 continued on next page*

*Figure 3 continued*

in mock (*n* = 67 cells) and TANGO2-depleted (*n* = 75 cells) cells in five samples for each condition by an ImageJ macro developed in our laboratory (open code access in Materials and methods). (**C**) Total volume of LDs present per cell. (**D**) Volume of each LD in a cell. (**E**) Number of LDs per cell. (**F, G**) Quantification of total LD volume in DMEM complete medium (control) and supplemented with 120 nM Oleic acid (+OA) using HCS LipidTOX Deep Red marker by flow cytometry. (**F**) Mean fluorescence intensity (mean FI) of allophycocyanin (APC) detected in Mock and TANGO2-depleted HepG2 cells in three different samples of each condition. (**G**) Mean fluorescence intensity (mean FI) of APC detected in wild-type (WT) and TANGO2-deficient disease (TAN043) fibroblasts in three different samples of each condition. (**H**) Quantification of total LD's volume in DMEM complete medium (control), supplemented with 120 nM Oleic acid (+OA), or supplemented with 120 nM OA and 500 µM diethylumbelliferyl phosphate (DEUP) in WT and TAN043 fibroblasts by flow cytometry. (**I**) Quantification of total LD volume after supplementation with 240 nM OA for 12 hr, washed and starved for 5 hr with DEUP in WT and TAN043 fibroblasts by flow cytometry. Images and graphs are representatives of three independent experiments. In graphs, boxes and bars are the mean ± standard deviation (SD). A.U. means arbitrary units.

The online version of this article includes the following figure supplement(s) for figure 3:

**Figure supplement 1.** Depletion of TANGO2 in HepG2 cells.

complete-medium conditions revealed a greater increase of LipidTOX marker in TANGO2-depleted fibroblasts compared to WT (*Figure 3H*). This result suggests a more prominent activation of the LD biogenesis in TANGO2-lacking fibroblasts. To test the effect on LD consumption, we added 240 nM OA at 37°C for 12 hr in DMEM complete medium. We then changed the supplemented OA medium with fresh-complete medium including DEUP to synchronize lipolysis initiation (+OA + DEUP), nutrient-free medium to activate starvation (+OA + Starvation) or nutrient-free medium with DEUP to block lipolysis (+OA + Starvation + DEUP) (*Figure 3—figure supplement 1E*). Fasting induction demonstrated a drastic alteration of the lipolytic pathway in TANGO2-depleted fibroblasts compared to WT fibroblasts (*Figure 3I*).

Altogether, depletion of TANGO2 displays substantial enlargement of LDs by a combination of increased biogenesis and a pronounced defect in lipolysis.

## TANGO2 depletion affects the quantities of specific lipids

A change in the size of LDs in TANGO2-depleted cells prompted us to analyze changes in the cellular lipid composition. Also of note is the observation that patients with TANGO2 deficiency suffer metabolic crises upon fasting, which suggests an impairment in lipid metabolism. Mock cells (Mock), TANGO2-depleted cells (siTANGO2), starved mock cells (Starvation), and starved TANGO2-depleted cells (siTANGO2 starvation) were analyzed in triplicates to quantitate their total lipid composition (Lipotype GmbH). We used the mole percent fraction (molp) analyses because that enables comparisons between the lipidomic profiles of different conditions, disregarding the absolute amount of lipids (*Figure 3—figure supplement 1A*). However, we also report the absolute levels of lipids in our conditions (*Figure 3—figure supplement 1B, C*).

First, to monitor the effect of physiological starvation, we compared samples of mock cells under control and starvation conditions (*Figure 4*, left panel). Upon starvation, LPA levels are reduced, whereas phosphatidylcholine (PC), lysophosphoglycerol (LPG), phosphoglycerol (PG), and PA levels are notably increased (*Figure 4*, left panel). LPA is used to produce PA, which is a precursor of PG and mitochondria-specific CL.

Secondly, to monitor the effect TANGO2 depletion, we compared samples of Mock and TANGO2-depleted cells grown in normal condition (*Figure 4*, middle panel). TANGO2-depleted cells show an increase in levels of LPA and reduction in PA and CL levels. There was also an increase in most lysophospholipids and a reduction in the corresponding phospholipids (*Figure 4*, middle panel).

Lastly, to monitor the combine effect of starvation and TANGO2 depletion, we compared starved TANGO2-depleted cells with mock starved cells (*Figure 4*, right panel). Importantly, starving TANGO2-depleted cells exhibited a further increase in LPA and decrease in PA levels (*Figure 4*, right panel). Moreover, in TANGO2-depleted cells, LPG levels are reduced and there is a dramatic reduction in PG levels. The levels of these two lipids are further reduced in early starvation. We suggest that when PA levels are low, the cells use LPG and PG to produce CL in starving TANGO2-depleted cells and this is the likely reason for the partial restoration of CL levels under these conditions. These data show that nutrient-starved TANGO2-depleted cells have a dramatic change in their LPA to PA ratio and there is a subsequent defect in the homeostasis of additional fatty acids.

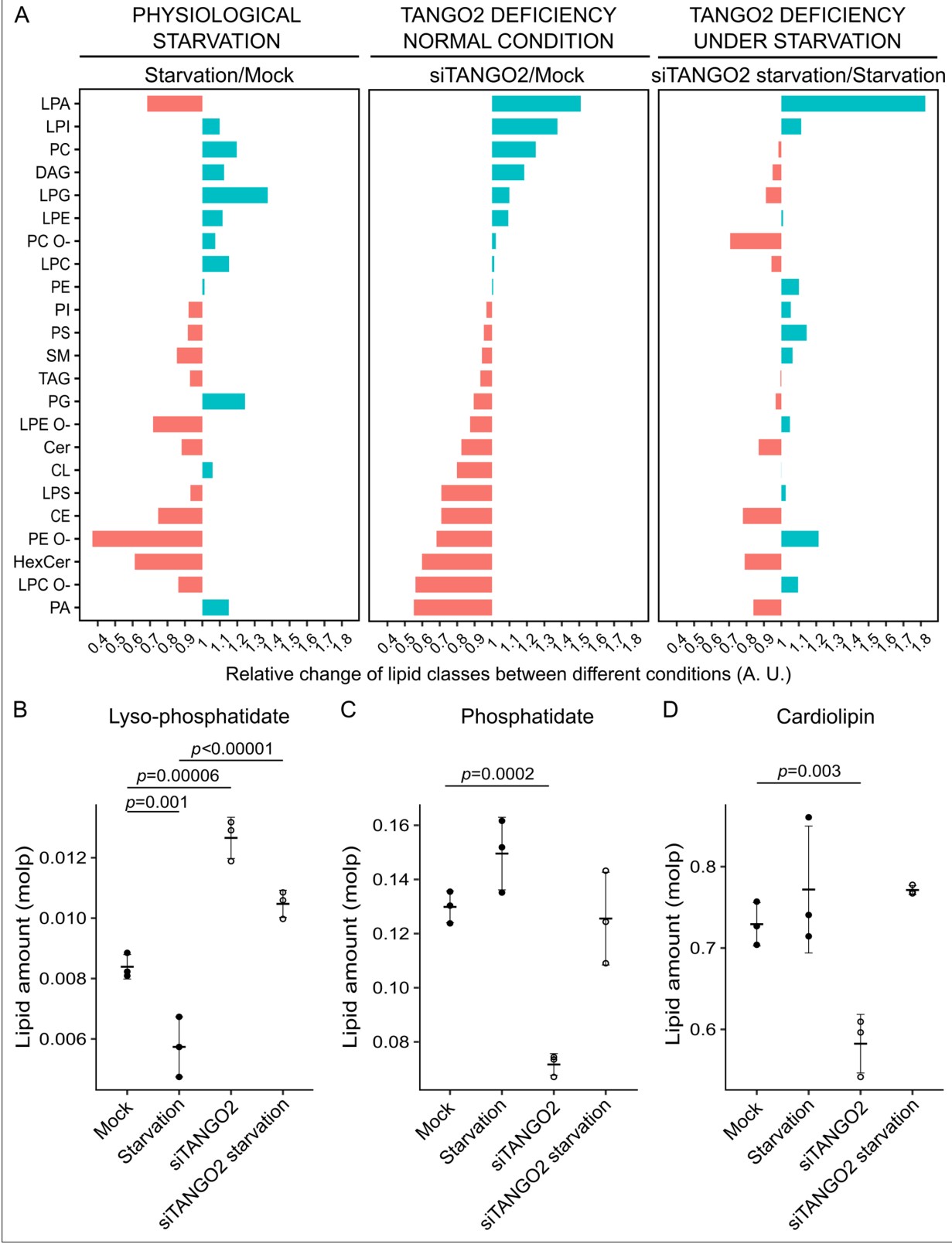

**Figure 4.** Lipidomics of TANGO2-depleted cells for comparison with control and nutrient-starved cells. (**A**) Relative change of lipid class according to different conditions: physiological starvation (left panel), TANGO2-depleted cells in control conditions (middle panel), and TANGO2-depleted cells upon nutrient starvation (right panel). (**B**) Quantification of lysophosphatidic acid (LPA) changes in mock and TANGO2-depleted cells expressed in mole percent (molp) fractions. (**C**) Quantification of phosphatidic acid (PA) changes in mock and TANGO2-depleted cells expressed in molp fractions.

*Figure 4 continued on next page*

*Figure 4 continued*

(**D**) Quantification of CL changes in mock and TANGO2-depleted cells expressed in molp fractions. Data are representative of three independent experiments. In graphs, boxes are the mean ± standard deviation (SD).

The online version of this article includes the following figure supplement(s) for figure 4:

**Figure supplement 1.** Absolute values comparison of lipids in HepG2 mock and TANGO2-depleted cells.

## The loss of TANGO2 affects the levels of enzymes involved in the metabolism of phospholipids and neutral lipids

Changes in the lipid profiles of TANGO2-depleted cells led us to analyze enzymes involved in lipid metabolism. PA has a central role as precursor of all components synthesized in the phospholipid biosynthetic pathway. In short, the glycerol-3-phosphate acyltransferases (GPATs) located in the ER (GPAT3/4) and OMM (GPAT1/2) membranes, catalyze the conversion of glycerol-3-phosphate (G3P) to LPA, by transferring a saturated fatty acid from Acyl-CoA. Then, the family of 1-acyl-sn-glycerol-3-phosphate acyltransferases (AGPATs), with lysophosphatidic acid acyltransferase (LPAAT) enzymatic activity, promote addition of an unsaturated fatty acid from acyl-CoA to LPA to form PA. Once formed, PA can generate diacylglycerol (DAG) after dephosphorylation by PA phosphatase enzymes (LPINs), or can reconvert to LPA by the action of phospholipase A2 (PLA2).

The expression of 13 enzymes involved in the phospholipid biosynthetic pathway was measured in three independent experiments by reverse transcription quantitative polymerase chain reaction (RT-qPCR). RNA extraction was performed in Mock and TANGO2-depleted HepG2 cells in complete medium and starvation medium. For mRNA quantification, we used GAPDH as a reference gene in all conditions. RT-qPCR graphs showed the normalized mRNA expression of each enzyme in TANGO2-depleted cells compared to Mock cells in three independent experiments. These results revealed an increase in GPAT3 and AGPAT1 mRNA expression inTANGO2-depleted cells compared to Mock cells in both control and fasting conditions (*Figure 5A, B*). Interestingly, AGPAT3 mRNA expression was decreased in control conditions but significantly increased in a nutrient-free conditioned medium. Furthermore, AGPAT5 mRNA expression was significantly higher in control conditions but not in nutrient deprivation conditions. However, the opposite was seen for AGPAT2 mRNA expression. The overexpression of AGPAT1 and AGPAT3 enzymes should increase PA production, but TANGO2 deficiency showed a decrease in the PA lipid profile by lipidomic analysis (*Figure 4*). Moreover, the increase in GPAT3 mRNA expression might contribute to LPA formation.

Although further enzymatic studies should explore their location and functional activity, these data suggest that decreased production of PA triggers mechanisms to re-establish the physiological levels of this essential component, but the lack of TANGO2 prevents this restorative mechanism.

## TANGO2-depleted cells exhibit increased ROS and lipid peroxidation

Given the location of TANGO2 to mitochondria and a previous report of increased mitochondria ROS in cells of patients with a TANGO2 deficiency (*Heiman et al., 2022*), prompted us to test this feature in our experimental conditions. Mitochondria produce ROS at the inner mitochondria membrane during oxidative phosphorylation. ROS produced as a byproduct of ATP synthesis is useful for cells, but high levels of ROS, unless scavenged, are toxic. Mock HepG2 cells (Mock), TANGO2 transiently depleted cells (siTANGO2), wild-type human fibroblasts (WT), and TANGO2 knock-out cells from TDD patients (TAN043) in complete medium or starvation medium, were incubated for 20 min with CellROX Deep Red reagent to measure cellular ROS levels and MitoTracker Green to measure total mitochondria mass by flow cytometry analysis. Cell viability was measured by using CellTrace Calcein Green AM. DNA-specific DAPI was used to test that TANGO2 silencing did not induce cell death in our samples (*Figure 6—figure supplement 1*). To determine the total mitochondria mass, we analyzed the fluorescence intensity (FI) of fluorescein isothiocyanate (FITC) corresponding to MitoTracker Green for all samples by flow cytometry (*Figure 6A*). The statistical analysis shows the mean FI (MFI) of FITC in three independent experiments. The results reveal no significant difference in total mitochondria mass in TANGO2-depleted cells compared to controls (*Figure 6B*). To measure ROS production in these experimental conditions, we analyzed differences in FI of APC corresponding to CellROX marker (*Figure 6C*). The data show that depletion of TANGO2 significantly increased ROS levels, which are further elevated upon nutrient starvation (*Figure 6D*). As we observed a significant difference in the

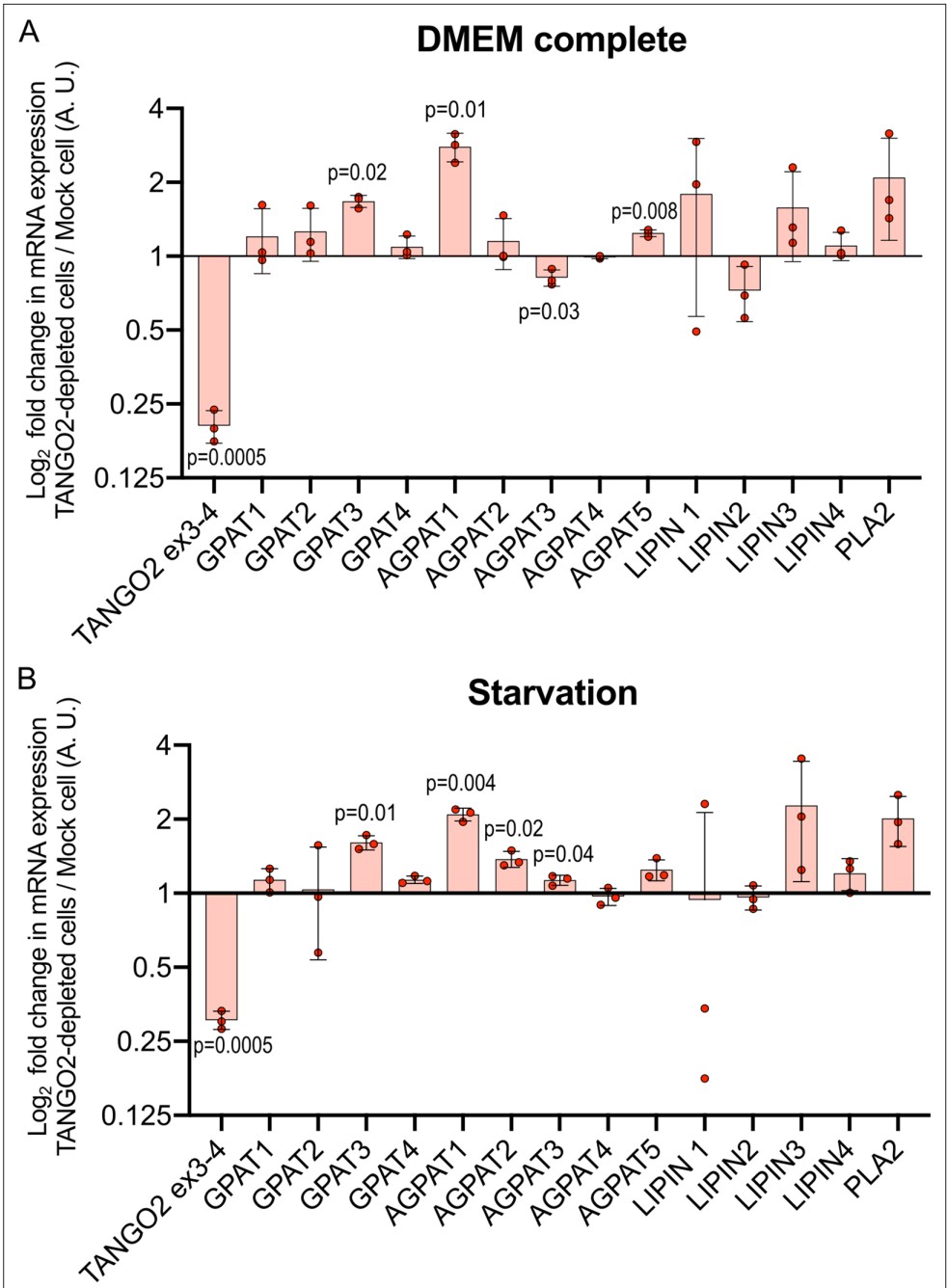

**Figure 5.** TANGO2 depletion impacts enzymes of lipid metabolism. Reverse transcription quantitative polymerase chain reaction (RT-qPCR) analyses of TANGO2-depleted HepG2 cells compared to mock cells in normal (**A**) and starvation (**B**) conditions. Three biological repeats were performed for each condition. Values were normalized to GAPDH and graphed as log fold change relative to Mock cells of each condition. Statistical significance was determined using a simple unpaired Student's t-test with subsequent Welch's correction analysis between TANGO2-depleted and Mock cells. In graphs, boxes are the mean ± standard deviation (SD). A.U. means arbitrary units.

mitochondria mass between fibroblasts from healthy and TDD donors, we analyzed ROS levels by the ratio between CellROX and MitoTracker markers. These results corroborate that TANGO2-lacking fibroblasts have higher ROS levels than WT fibroblasts (**Figure 6E**).

Since the increase in ROS damage cellular components such as DNA, proteins, and lipids, we analyzed the peroxidation state of lipids. HepG2 cells transiently depleted of TANGO2 (Mock vs.

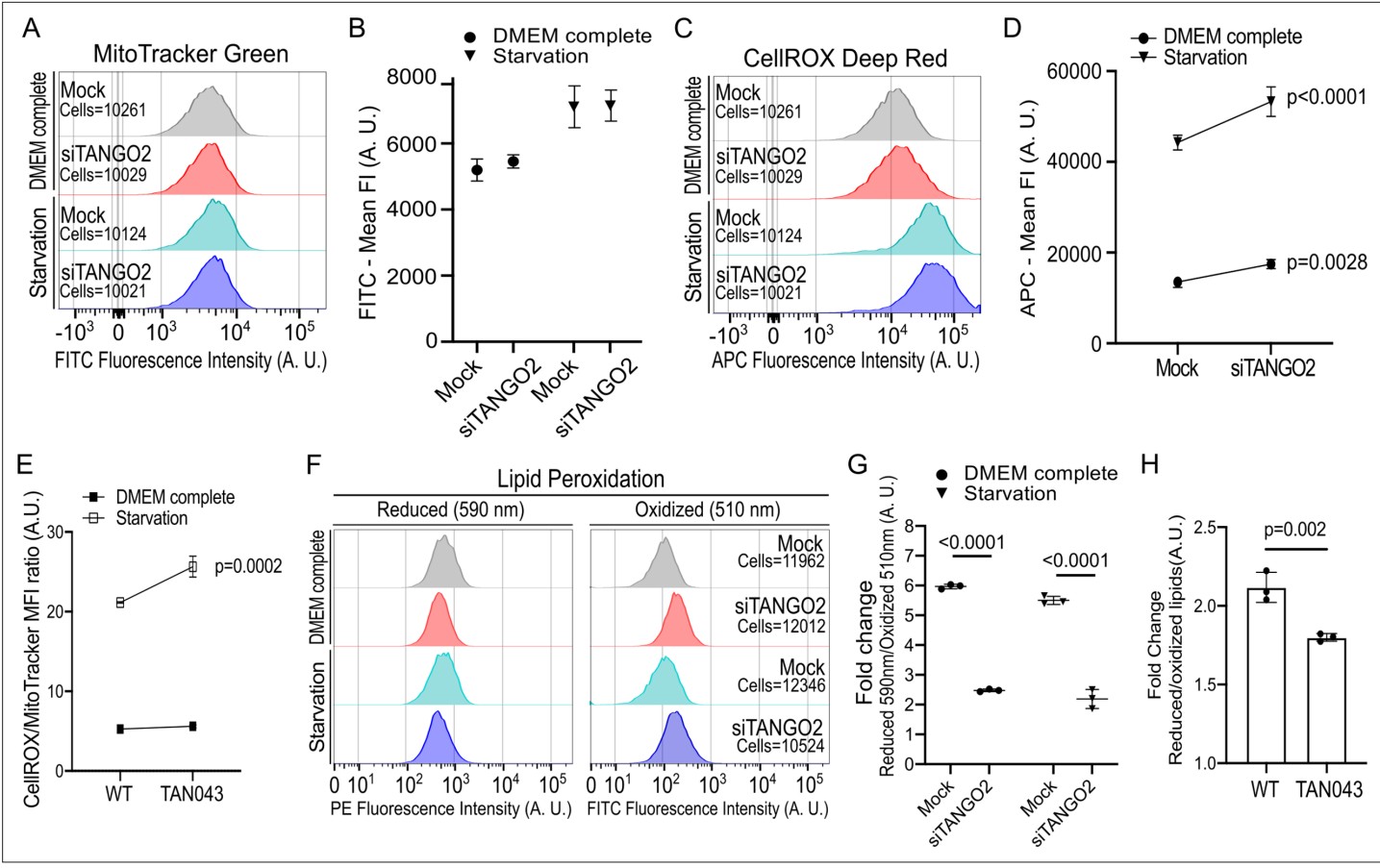

**Figure 6.** TANGO2-depleted cells exhibit increased reactive oxygen species (ROS) levels and lipid peroxidation. (**A–D**) Mock and TANGO2-depleted cells were incubated in control conditions (Dulbecco's Modified Eagle's Medium [DMEM] complete) or in a low-nutrient medium (Starvation) for 4 hr. Cells were incubated with CellROX Deep Red to detect ROS levels and MitoTracker Green to monitor mitochondria mass. (**A**) Mitochondria mass was detected by measuring the fluorescence intensity of fluorescein isothiocyanate (FITC) by flow cytometry. (**B**) The mean fluorescence intensity (Mean FI) of FITC was measured in three different samples of each condition. (**C**) Cellular ROS levels were detected by measuring the fluorescence intensity of allophycocyanin (APC) by flow cytometry. (**D**) The mean fluorescence intensity of APC was measured in three different samples of each condition. (**E**) Ratio between ROS levels (CellROX) and mitochondria mass (MitoTracker) was measured in WT and TAN043 fibroblasts by flow cytometry in three independent experiments. (**F**) Lipid peroxidation in HepG2 and fibroblasts was measured by detecting the mean fluorescence intensity change between PE and FITC spectral emissions by flow cytometry. (**G**) The fold change shows the ratio between reduced and oxidized lipid species in Mock and TANGO2-depleted HepG2 cells in the control and nutrient-free medium. (**H**) The fold change shows the ratio between reduced and oxidized lipid species in WT and TAN043 fibroblasts. In graphs, bars and boxes are the mean ± standard deviation (SD). A.U. means arbitrary units.

The online version of this article includes the following figure supplement(s) for figure 6:

**Figure supplement 1.** Cell viability of TANGO2-depleted cells.

siTANGO2) and fibroblasts derived from TDD donors (WT vs. TAN043) were incubated for 30 min with an Image-iT Lipid Peroxidation kit, a sensitive fluorescent reporter for measuring total lipid peroxidation, by flow cytometry. In a reduced state, the maximal emission of the sensor is 591 nm, corresponding to the FI of Phycoerythrin (PE) (*Figure 6F*, left panel). Upon oxidation, the reagent shifts the fluorescence emission peak from ~590 to ~510 nm corresponding to the FI of FITC (*Figure 6F*, right panel). To determine lipid peroxidation, we analyzed the MFI ratio between PE and FITC for all samples (*Figure 6G, H*). The statistical analysis shows the fold change of reduced/oxidized lipids in three independent experiments. The results show that rise in ROS production provokes an increase in lipid peroxidation of TANGO2-depleted HepG2 cells (*Figure 6G*) and TAN043 human fibroblasts (*Figure 6H*).

Altogether, these data show that TANGO2 depletion causes unsaturated lipid peroxidation, which is known to alter lipid quality.

## Discussion

TANGO2, a less studied protein from the TANGO collection, is gaining attention recently because of mutations in TANGO2 lead to severe pathologies (*Bérat et al., 2021*; *Hoebeke et al., 2021*; *Powell et al., 2021*; *Schymick et al., 2022*). Patients with mutations in TANGO2 exhibit defects in cardiac and neuronal physiology leading to fatalities upon starvation and stress (https://tango2research.org; *Lalani et al., 2016*; *Miyake et al., 2022*). Here, we describe the physiological role of TANGO2 and its link to starvation-induced pathologies in patients.

### TANGO2 functions in lipid homeostasis

TANGO2 has been shown to function at the mitochondria and in the ER-to-Golgi step of the protein secretion pathway (*Milev et al., 2021*; *Bard et al., 2006*). Our new data reveal that TANGO2 localizes predominantly to mitochondria, but also at sites on mitochondria that are juxtaposed to LDs, and ER (*Figure 2* and *Figure 2—figure supplement 1*).

Loss of TANGO2 in HepG2 cells has revealed four clear features: (1) a change in the abundance of several phospholipids, mainly in the ratio of LPA to PA. LPA levels are elevated, and PA is reduced; (2) reduction in CL content; (3) increase in ROS and unsaturated lipid peroxidation levels; (4) increase in LD size. The significance of these changes to TANGO2 function in physiology and pathology follows.

1. LPA is acylated to generate PA by lipid-modifying enzymes. The fact that LPA increases and PA decreases suggests that acylation of LPA is defective in TANGO2-lacking cells. The enzymes (LPAATs) involved in these acylation reactions are transmembrane proteins and acyl-CoA produced in the cytosol has to be imported into the lumen by membrane-embedded transporters (*Gonzalez-Baro and Coleman, 2017*; *Lu and Claypool, 2015*). TANGO2 lacks a transmembrane domain and cannot therefore function as an enzyme to catalyze these reactions in the lumen or the membrane of the ER or mitochondria. We suggest TANGO2 likely functions on the cytoplasmic face of mitochondria and the ER in the synthesis of acyl-CoA, or a carrier protein for acyl-CoA collection, or its metabolism. Our data show that although predominantly located to mitochondria, there is an enrichment of TANGO2 at sites juxtaposed to ER. This could be the site of LPA to PA conversion (*Gonzalez-Baro and Coleman, 2017*; *Lu and Claypool, 2015*). We suggest that without TANGO2, acyl-CoA is unavailable for the synthesis of PA from LPA. As a result, LPA levels rise with a concomitant drop in PA. This effect is exacerbated in nutrient-starved TANGO2-depleted cells. We noticed an increase in the expression of GPAT and AGPAT in cells depleted of TANGO2. This suggests that cells sense a reduction in the levels of PA and upregulated these enzymes that convert G3P to LPA and LPA to PA. Why do not we see restoration of PA levels in TANGO2-depleted cells. There are two possibilities: (1) TANGO2 is necessary to collect acyl-CoA at the membrane and in its absence the cells do not have the pool of acyl-CoA available for producing PA. (2) In cells lacking TANGO2, there is an increase in ROS production, which modifies unsaturated fatty acids, which cannot be incorporated into PA. These possibilities remain untested.

2. A reduction in CL at the inner mitochondria membrane (*Falabella et al., 2021*; *Paradies et al., 2019*) is likely due to reduced LPA acylation because PA is a precursor to CL. Reduced CL in TANGO2-depleted cells could explain the consequent effects on mitochondria morphology and physiology (*Heiman et al., 2022*). In starving TANGO2-depleted cells, CL levels recover slightly compared to TANGO2-depleted cells. We suggest that when PA is low, cells likely use PG to produce CL (*Falabella et al., 2021*). This explains reduction in PG levels in starving TANGO2-depleted cells compared. Changes in CL are associated with a number of cardiac diseases (*Dudek, 2017*; *Mulligan et al., 2014*; *Petrosillo et al., 2005*). For example, Barth syndrome (BTHS) is an inherited cardiomyopathy, associated with skeletal myopathy and growth retardation (*Barth et al., 1996*) , caused by a mutation in a mitochondria phospholipid-lysophospholipidtransacylase, involved in the biogenesis of CL (*Dudek and Maack, 2017*). These features are similar to the pathologies associated with TANGO2 mutations in humans.

3. Raising ROS levels cause mitochondria dysfunction (*Bhatti et al., 2017*; *Sies and Jones, 2020*). These oxidative modifications could alter the lipolytic mechanisms necessary to meet ATP demands during fasting. Moreover, high ROS levels cause peroxidation of unsaturated lipids that could be required in the LPA to PA step. Also, lipid peroxidation affects intracellular membranes, which could explain the effect of protein targeting and secretion (*Sies and Jones, 2020*).

4. Changes in LD volume are probably a reflection of increased incorporation to peroxidized lipids and a defect in catabolism. Several recent publications suggest that LD expansion upon

oxidative stress is an evolutionarily conserved phenomenon to meliorated ROS-mediated lipid damage (*Gasparian et al., 2022*; *Herms et al., 2013*; *Liu et al., 2015*; *Rambold et al., 2015*). As a result, there is a defect in the availability of lipids for ATP production, mitochondria homeostasis, and cell signaling.

Altogether, TANGO2-depleted cells exhibit a change in lipid metabolism, which is aggravated in starving cells.

## TANGO2 as a haem transfer protein

A paper published recently by Chen et al. showed that loss of human TANGO2 and its orthologs in yeast and worms caused accumulation of haem in compartments like the mitochondria and lysosome-related organelles (LROs) (*Sun et al., 2022*). The authors showed that recombinant TANGO2 has low affinity binding to ferrous and ferric haem and adding recombinant TANGO2 to mitochondria enriched membrane fractions in vitro stimulated haem transfer to apo-myoglobin. They have concluded that haem is exported from the lumen into the cytoplasm by transmembrane transporters where it is captured by TANGO2 and subsequently transferred to specific clients. Surprisingly, this was observed even in trypsinized mitochondria and the authors concluded that cytoplasmic facing domains of trans-membrane proteins or attached proteins are not required in TANGO2-mediated haem transfer. Why do TANGO2-depleted cells accumulate haem in mitochondria (or LROs)? If TANGO2 captures haem post its export from mitochondria, as is reported even in trypsinized mitochondria, then haem should be distributed into the cytoplasm and not retained in mitochondria or LROs. The data also do not explain the aggravated effects of TANGO2 depletion linked to human pathologies, including a lack of any obvious sign of anaemia as would be evident for defects in haem physiology. A plausible explanation for the data is that amphipathic haem is retained in mitochondria because of altered membrane lipid composition upon TANGO2 depletion. Further work is therefore required to clarify whether the proposed function of TANGO2 in haem transfer from mitochondria is an indirect consequence of dysfunctional mitochondria lipid composition.

## Conclusion

We propose that TANGO2 has a function in events leading to specific phospholipid production. Our data show a significant change in the ratio of LPA to PA in TANGO2-depleted cells. A defect in PA levels impacts the synthesis of mitochondria CL, and to balance this deficiency, the cells use other potential substrates, such as the PG, to produce CL. The net effect is an imbalance in lipid composition or homeostasis that, we suggest, also increases ROS levels and promotes lipid peroxidation. The acyl-CoA pathway is also used for the synthesis of numerous fatty acids, and a defect in its production would change the form and function of LDs to influence cellular lipid homeostasis. Sacher et al. have reconstituted many features of TANGO2 mutations in flies (*Asadi et al., 2022*). Interestingly, most of the aberrations in fly physiology are corrected, and an ER-to-Golgi transport defect in human cells by vitamin B5 supplement, which is used for the synthesis of CoA. In other words, loss of TANGO2 function is restored by replenishing or overdosing events leading to lipid neogenesis. CoA is used for acyl-CoA production, which is then used to acylate LPA to form PA. It has recently been shown that B-complex and multivitamins might prevent metabolic crises (*Asadi et al., 2022*; *Miyake et al., 2022*; *Sandkuhler et al., 2023*). However, why do multivitamins ameliorate metabolic crises in TANGO2 patients? Based on our results, we suggest that it is likely a combined effect of multivitamins on their antioxidant properties and in stimulating new lipids biosynthesis. The starving cells rely heavily on lipids for ATP production, and TANGO2 depleted are therefore further stressed. Our findings thus reveal the physiological role of TANGO2 and provide a handle to understand how its functional loss triggers a variety of cardiac, muscular, and neurological pathologies leading to fatalities upon conditions of starvation and stress.

## Materials and methods
### Cell culture
HepG2 cells (ATCC, Cat# HB-8065) were cultured in DMEM (Lonza, Cat# SH30243.01) cell culture media supplemented with 10% (vol/vol) heat-inactivated FBS (Gibco, Cat# 10270-106) (DMEM complete), 100 units/ml penicillin, and 100 µg/ml streptomycin (Gibco), at 37°C in a humidified

incubator supplied with 5% $CO_2$. Primary dermal fibroblasts were isolated from healthy donors (WT) and TDD patients (exon 3–9 deletion mutation in TANGO2 gene; TAN043). Human skin fibroblasts were obtained with the patient's consent using the BCM IRB Protocol H-43240 (Clinical Trials Identifier: NCT05374616, Baylor College of Medicine, Houston, TX, USA). Fibroblasts were cultured in DMEM medium supplemented with 20% (vol/vol) heat-inactivated FBS (FBS; Gibco, Cat# 10270-106), 100 units/ml penicillin, and 100 µg/ml streptomycin (Gibco) at 37°C in a humidified incubator supplied with 5% $CO_2$. The cells used in our experiments were negative for the mycoplasma test. For starvation conditions, cells were incubated in OptiMEM (Gibco, Cat# 11058021) without FBS for 4 hr.

## Plasmids, RNA interference, and cell transfection

For transient protein overexpression, cells were plated and 24 hr later transfected with 1:1 ratio (DNA plasmid:Lipofectamine) using Lipofectamine 3000 reagent (Thermo Fisher Scientific) following the manufacturer's recommendations. Plasmids generated in our laboratory: TANGO2.Isoform1-EGFP, TANGO2.Isoform1-mScarlet, Peroxisome-SKL-mTurquoise2, and GPAT4-Hairpin-mScarlet. Plasmids used and available from Addgene: ER-pmTurquoise2 (Addgene, Cat# 36204) and Mitochondria-pmTurquoise2 (Addgene, Cat# 36208). For TANGO2 silencing, cells were plated and transfected with 1:3 ratio (RNA:Lipofectamine) using Lipofectamine RNAiMAX reagent (Thermo Fisher Scientific) following the manufacturer's recommendations. Twenty-four hours after the first transfection, cells were transfected with a second and identical mix (ratio 1:3) of RNA:Lipofectamine. Two days after the second transfection, experiments described in this paper with siTANGO2 conditions were performed. TANGO2 RNAi sequences: Hs.Ri.TANGO2.13.1-SEQ1 (IDT technologies, Mfg ID# M498082302) and Hs.Ri.TANGO2.13.1-SEQ2 (IDT technologies, Mfg ID# M498082303).

## Antibodies and reagents

Primary antibodies used: anti-ADRP (Santa Cruz, Cat# sc-377429), anti-GFP (Roche, Cat# 11814460001), anti-β-actin (Sigma, Cat# A1978), anti-β-tubulin (Sigma, Cat# T4026), and anti-Tom20 (Santa Cruz, Cat# sc-17764). Secondary antibodies used: Alexa Fluor 594 donkey anti-rabbit (Invitrogen, Cat# A21207), Alexa Fluor 594 donkey anti-mouse (Invitrogen, Cat# A21203), Alexa Fluor 633 donkey anti-mouse (Invitrogen, Cat# A21050), Alexa Fluor 647 donkey anti-rabbit (Invitrogen, Cat# A31573), Alexa Fluor 647 donkey anti-mouse (Invitrogen, Cat# A31571), Alexa Fluor 680 donkey anti-rabbit (Invitrogen, Cat# A10043), and Alexa Fluor 800 donkey anti-mouse (Invitrogen, Cat# A32789). Reagents used were: DAPI (Invitrogen, Cat# B223), Hoechst-33342 (Sigma, Cat# B2261), HCS LipidTOX Deep Red (Thermo Fisher Scientific, Cat# H34477), Mitotracker Green (Thermo Fisher Scientific, Cat# M7514), CellROX Deep Red (Thermo Fisher Scientific, Cat# C10422), Image-iT Lipid Peroxidation Kit (Thermo Fisher Scientific, Cat# C10445), CellTrace Calcein Green, AM (Thermo Fisher Scientific, Cat# C34852), FluorSave reagent (Millipore, Cat# 345789-20), Lipofectamine RNAiMAX reagent (Thermo Fisher Scientific, Cat# 13778075), and Lipofectamine 3000 reagent (ThermoFisher Scientific, Cat# L3000015).

## Immunoblotting

Immunoblot analysis was performed following the Biorad general protocol recommendations. Briefly, equal amounts of protein were resolved by SDS–PAGE, transferred onto 0.45-µm polyvinylidene difluoride (PVDF) membranes (Amersham), and incubated overnight with anti-GFP (1:1000; Abcam), anti-β-actin (1:10,000; Sigma) antibodies followed by Alexa-conjugated secondary antibodies (1:20,000; Invitrogen) for 1 hr at 37°C. Bands were visualized an Odyssey CLX (LI-COR Biosciences).

## Immunofluorescence and immunohistochemistry

HepG2 cells (1 × $10^5$ cells per well) were grown on coverslips. After mock or siTANGO2 silencing, cells were fixed with 3% paraformaldehyde, quenched, and incubated overnight with anti-ADRP (1:1000) at 4°C followed by Alexa Fluor 647 donkey anti-mouse (1:2000) and DAPI (1:10,000) for 1 hr at 37°C. Coverslips were washed and mounted in FluorSave reagent. Samples were analyzed and high-resolution images were acquired in an inverted Leica TCS SP8 confocal microscope equipped with photomultipliers and hybrid detectors. Images were processed using ImageJ software.

## Förster resonance energy transfer measurement

For the energy donor and acceptor pair we selected NeonGreen and mScarlet, respectively, as they have been previously described as a suitable Förster resonance energy transfer (FRET) pair (*McCullock*

*et al., 2020*). To measure the energy transfer between the two we used the 'acceptor photobleaching' method for its simplicity and robustness as has been previously described (*Grecco and Bastiaens, 2013*). By comparing the change in intensity of the quenched (pre-photobleaching) and unquenched (post-photobleaching) images of the donor we can measure FRET between TANGO2mScarlet and Tom20-Alexa 633 using the following formula: FRET efficiency = 1 − (quenched donor/unquenched donor). To calculate the FRET efficiency we used averaged intensity values of each image. To calculate FRET maps we used pixel by pixel calculation of the FRET efficiency. All image calculations were processed using ImageJ software.

### Live-cell imaging

HepG2 cells ($2.5 \times 10^5$ cells per dish) were grown on 35-mm polymer-bottom dishes (ibidi). After mock or siTANGO2 silencing, cells were washed and the cell medium was replaced with the appropriate conditioned medium for the indicated period of time. The cells were then washed several times to remove the conditioned medium and incubated with the appropriate reagent (Hoechst-33342, HCS LipiTOX Deep Red, MitoTracker Green). Finally, after several washes, cells were used to perform the experiments described in the paper. Samples were analyzed at 37°C in 5% $CO_2$ atmosphere. High-resolution and time-lapse images were acquired in an inverted Leica STELLARIS confocal microscope equipped with photomultipliers and hybrid detectors. Images were processed using ImageJ software. For co-localization analysis, we calculated the Pearson correlation coefficient ($r$) of confocal images with the coloc2 plugin in ImageJ software. To quantify the volume of LDs in confocal images, we developed a macro to identify each voxel ($pixel^3$) stained by the lipophilic marker LipidTOX in $z$-stacks. We then selected five $z$-stacks from each condition. This macro automatically identified and distinguished the positive from the negative particles on each slide and generated a data set with the volume of each particle, the number of particles, and the total volume of positive particles. This method reduced the chances in counting errors or biases.

### Flow cytometry assays

To examine the total LDs volume, HepG2 cells ($2 \times 10^5$ cells per well) were seeded in 24-well plates (Thermo Fisher Scientific). Two days after the second silencing (Mock or siTANGO2), cells were washed and the cell medium was changed with the appropriate medium for the times indicated. The cells were then washed several times to remove the conditioned medium and incubated with the appropriate reagent (CellROX Deep Red, Lipid Peroxidation Red/Green, HCS LipidTOX Deep Red, MitoTracker Green, Calcein AM, DAPI). Cells were gently detached using trypsin and washed several times before flow cytometry analyses. Flow cytometry assays were performed by technical triplicates of at least 10,000 cells per condition each time in three independent experiments. In the statistical model, these triplicates are nested together. Flow cytometry analysis was performed using a BD LSR II Flow Cytometer (Becton Dickinson Biosciences).

### Lipid extraction for mass spectrometry lipidomics

Mass spectrometry-based lipid analysis was performed by Lipotype GmbH (Dresden, Germany) as described (*Surma et al., 2021*). Lipids were extracted using a chloroform/methanol procedure (*Ejsing et al., 2009*). Samples were spiked with internal lipid standard mixture containing: cardiolipin 14:0/14:0/14:0/14:0 (CL), ceramide 18:1;2/17:0 (Cer), diacylglycerol 17:0/17:0 (DAG), hexosylceramide 18:1;2/12:0 (HexCer), lyso-phosphatidate 17:0 (LPA), lyso-phosphatidylcholine 12:0 (LPC), lyso-phosphatidylethanolamine 17:1 (LPE), lyso-phosphatidylglycerol 17:1 (LPG), lyso-phosphatidylinositol 17:1 (LPI), lyso-phosphatidylserine 17:1 (LPS), phosphatidate 17:0/17:0 (PA), phosphatidylcholine 17:0/17:0 (PC), phosphatidylethanolamine 17:0/17:0 (PE), phosphatidylglycerol 17:0/17:0 (PG), phosphatidylinositol 16:0/16:0 (PI), phosphatidylserine 17:0/17:0 (PS), cholesterol ester 20:0 (CE), sphingomyelin 18:1;2/12:0;0 (SM), and triacylglycerol 17:0/17:0/17:0 (TAG). After extraction, the organic phase was transferred to an infusion plate and dried in a speed vacuum concentrator. The dry extract was re-suspended in 7.5 mM ammonium formiate in chloroform/methanol/propanol (1:2:4, vol:vol:vol). All liquid handling steps were performed using Hamilton Robotics STARlet robotic platform with the Anti Droplet Control feature for organic solvents pipetting.

## MS data acquisition

Samples were analyzed by direct infusion on a QExactive mass spectrometer (Thermo Scientific) equipped with a TriVersa NanoMate ion source (Advion Biosciences). Samples were analyzed in both positive and negative ion modes with a resolution of Rm/z = 200 = 280,000 for MS and Rm/z = 200 = 17,500 for tandem mass spectrometry (MSMS) experiments, in a single acquisition. MSMS was triggered by an inclusion list encompassing corresponding MS mass ranges scanned in 1 Da increments (*Surma et al., 2015*). Both MS and MSMS data were combined to monitor CE, DAG, and TAG ions as ammonium adducts; LPC, LPC O-, PC, and PC O-, as formate adducts; and CL, LPS, PA, PE, PE O-, PG, PI, and PS as deprotonated anions. MS only was used to monitor LPA, LPE, LPE O-, LPG, and LPI as deprotonated anions; Cer, HexCer, and SM as formate adducts.

## Lipidomics data analysis and post-processing

Data were analyzed with an in-house developed lipid identification software based on LipidXplorer (*Herzog et al., 2012*). Data post-processing and normalization were performed using an in-house developed data management system. Only lipid identifications with a signal-to-noise ratio >5, and a signal intensity fivefold higher than in corresponding blank samples were considered for further data analysis.

## RNA extraction and RT-qPCR

RNA was extracted using an RNeasy mini kit (QIAGEN, Hilden, Germany) following the manufacturer's instructions for direct cell lysis in a 12-well plate. Luna Universal One-Step RT-qPCR Kit (New England Biolabs, Ipswich, MA) was used for RT-qPCR reactions. Briefly, 10 µl reaction contained: 5 µl Reaction MasterMix, 0.5 µl EnzymeMix, 0.8 µl primers (10 µM, Fwd + Rv mixed), 1 µl of RNA (20 ng), and 2.7 µl water. RT-qPCR reactions were performed in duplicate or triplicate, and three biological repeats were analyzed for each condition. RT-qPCRs were performed in a LightCycler 480 System (Roche, Basel, Switzerland) with conditions as follows: 55°C for 10 min, 95°C for 1 min, followed by 45 cycles of 95°C for 10 s, 60°C for 30 s. The cycle threshold value (Ct) was calculated for each sample and normalized to GAPDH. The relative expression levels were calculated using ΔΔCT method. Primer sequences are included in the following table:

| Gene | Forward primer (5′–3′) | Reverse primer (5′–3′) |
|---|---|---|
| AGPAT1 | GGTACTCGCAACGACAATGG | TTGGTGTTGTAGAAGGAGGAGAAG |
| AGPAT2 | AACGTGGCGCCTTCCA | GAAGTCTTGGTAGGAGGACATGACT |
| AGPAT3 | CTCCAAGGTCCTCGCTAAGAAG | CCGCTTGCAGAACACAATCTC |
| AGPAT4 | CACGGAATGCACCATCTTCA | GAACCACGATGGCATTTTCCT |
| AGPAT5 | CTGGTGCTCCACACGTACTC | CCAGGCCAACACGTAGGT |
| GPAT1 | AACCCCAGTATCCCGTCTTT | CAGTCACATTGGTGGCAAAC |
| GPAT2 | GGCTGACGGAGGAGATACTG | AGTTGTGCCAGGTGTGTGAG |
| GPAT3 | ACAGCAGCCTCAAAAACTGG | CAATGGGGGAAGTATGGTTG |
| GPAT4 | TGCCAAATGGGAGGTTTAAG | GCCACCATTTCTTGGTCTGT |
| LPIN1 | CAGTCGAGGCTCAGACCA | TTCCCCGTTGATTTCTATGTCA |
| LPIN2 | CCTCTCCTCAGACCAGATCG | GGAGAATCTGTCCCAAAGCA |
| LPIN3 | AGAAGTCTTCACTGCAGCCC | CAGCTCCGAGTCGCTCTTAG |
| LPIN4 | CCTTCAGCCTGACACAGAGG | AAGCGCTGCATTCTCAGAGT |
| PLA2G4A | TCTACAACCCCTGACAGCAG | GCTGTCCCTAGAGTTTCATCCA |
| TANGO2 | CCGACCCTCCAAGTTAGCTG | TAGTTGGTGAGTGCTGCCAG |
| TANGO2 | CACAGCAAAGGGAGACGTCA | GTTCCACAGCCTCCAGGAAG |

## Bioinformatic analysis

The repository used in the in silico analysis of TANGO2 orthologs and isoforms is available at https://github.com/alessiovignoli/TANGO2 (copy archived at *Vignoli, 2023*).

The code used in the development of the macro to identify each voxel (voxel = pixel³ = pixel × pixel × pixel) is available at https://github.com/AAMateo/Lipid_Droplets (copy archived at *Farre, 2023*).

## Statistical analysis

Statistical analysis was performed using GraphPad Prism 8.0 (GraphPad) and R software. Data represent the mean ± standard deviations of *N* experiments. For simple unpaired analysis between two groups, Student's *t*-test with subsequent Welch's correction was choosen. For multiple comparisons, analysis of variance with subsequent Bonferroni's or Tukey's post-tests was used. p values less than 0.05 were considered statistically significant.

## Acknowledgements

We thank all members of the Malhotra laboratory for valuable discussions and critical reading of the manuscript. We thank Aida Rodriguez for advice with ROS analysis; Albert Pol and Albert Herms for advice with lipid metabolic experiments; the staff of the CRG/UPF Flow Cytometry Unit for advice with flow cytometry analysis and the staff of the CRG Advanced Light Microscopy Unit for invaluable technical help. We acknowledge the support of the Spanish Ministry of Science, the Centro de Excelencia Severo Ochoa, and the CERCA Programme/Generalitat de Catalunya. V Malhotra is an Institució Catalana de Recerca i Estudis Avançats professor at the Centre for Genomic Regulation. V Malhotra is an Institució Catalana de Recerca i Estudis Avançats professor at the Centre for Genomic Regulation Work in the Malhotra lab is funded by grants from the Spanish Ministry of Economy and Competitiveness (Plan Nacional to VM: PID2019-105518GB-I00) and the European Research Council Synergy Grant (ERC-2020-SyG-Proposal No. 951146). AL is funded by the European Molecular Biology Organization (EMBO ALTF 659-2021), JW is funded by the European Research Council (H2020-MSCA-IF-2019-894115). OF is funded by the Ramon y Cajal program (RYC-2016-20919). This work reflects only the authors' views, and the EU Community is not liable for any use that may be made of the information contained therein.

## Additional information

### Competing interests

Vivek Malhotra: Reviewing editor, eLife. The other authors declare that no competing interests exist.

### Funding

| Funder | Grant reference number | Author |
| --- | --- | --- |
| Ministerio de Asuntos Económicos y Transformación Digital, Gobierno de España | PID2019-105518GB-I00 | Vivek Malhotra |
| European Research Council | ERC-2020-SyG-Proposal No. 951146 | Vivek Malhotra |
| European Molecular Biology Organization | EMBO ALTF 659-2021 | Agustin Leonardo Lujan |
| European Research Council | H2020-MSCA-IF-2019-894115 | Jose Wojnacki |
| Ministerio de Ciencia e Innovación | RYC-2016-20919 | Ombretta Foresti |

The funders had no role in study design, data collection, and interpretation, or the decision to submit the work for publication.

## Author contributions
Agustin Leonardo Lujan, Conceptualization, Data curation, Formal analysis, Investigation, Methodology, Writing – original draft, Writing – review and editing; Ombretta Foresti, Conceptualization, Data curation, Formal analysis, Investigation, Methodology; Conor Sugden, Methodology; Nathalie Brouwers, Investigation, Methodology; Alex Mateo Farre, Alessio Vignoli, Software, Methodology; Mahshid Azamian, Alicia Turner, Resources; Jose Wojnacki, Data curation, Formal analysis, Investigation, Methodology, Writing – review and editing; Vivek Malhotra, Conceptualization, Data curation, Funding acquisition, Writing – original draft, Writing – review and editing

## Author ORCIDs
Agustin Leonardo Lujan ![ORCID] https://orcid.org/0000-0003-4906-6951
Ombretta Foresti ![ORCID] https://orcid.org/0000-0002-6878-0395
Nathalie Brouwers ![ORCID] https://orcid.org/0000-0002-9808-9394
Alessio Vignoli ![ORCID] https://orcid.org/0000-0001-7131-2915
Vivek Malhotra ![ORCID] http://orcid.org/0000-0001-6198-7943

## Decision letter and Author response
Decision letter https://doi.org/10.7554/eLife.85345.sa1
Author response https://doi.org/10.7554/eLife.85345.sa2

## Additional files

### Supplementary files
MDAR checklist

### Data availability
TANGO2 source data files are available at Zenodo (https://doi.org/10.5281/zenodo.19924215).

The following dataset was generated:

| Author(s) | Year | Dataset title | Dataset URL | Database and Identifier |
|---|---|---|---|---|
| Lujan AL | 2023 | Defects in lipid homeostasis reflect the function of TANGO2 in phospholipid and neutral lipid metabolism | https://doi.org/10.5281/zenodo.19924215 | Zenodo, 10.5281/zenodo.19924215 |

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
