## [Editor Report]

This important manuscript describes a series of cellular phenotypes associated with the depletion of TANGO2, a poorly characterized gene product but relevant to neurological and muscular disorders. The authors present solid data indicating that TANGO2 associates with membrane-bound organelles, mainly mitochondria, impacting lipid metabolism and the accumulation of reactive oxygen species.

---

## [Decision Letter]

**Decision letter after peer review:**

Thank you for submitting your article "Defects in lipid homeostasis reflect the function of TANGO2 in acyl-CoA metabolism" for consideration by *eLife*. Your article has been reviewed by 2 peer reviewers, and the evaluation has been overseen by a Reviewing Editor and Suzanne Pfeffer as the Senior Editor. The reviewers have opted to remain anonymous.

Essential revisions:

1. If possible, please analyze the localization of endogenous TANGO2 by immunofluorescence using the antibody described in Figure S2. The idea that TANGO2 localizes to membrane contact sites between mitochondria and the ER and LDs would also be strengthened by experiments including multiple organelle markers and/or methods to enhance the detection of membrane contact sites [https://doi.org/10.1073/pnas.1910854117].

2. Please display the absolute levels of all lipids in the various conditions.

3. RNAi-depletion of Tango2 causes LD accumulation, but it is unclear whether this is due to decreased LD lipolysis, increased LD biogenesis, or some combination of both. Pulse-chase 14C-palmitate experiments (or heavy label palmitate with mass spec) would substantially help the work and would also help determine if rates of acyl-CoA incorporation/turnover into phospholipids are defective in tango2-RNAi cells.

4. Check whether Tango2 loss impacts AGPAT, LPAAT, and Lipin enzyme levels and/or transcription.

*Reviewer #1 (Recommendations for the authors):*

Below are additional specific comments/suggestions that may help improve the manuscript.

1. The accumulation of lysoPA in Tango2-depleted cells may suggest a defect in AGPAT enzymes, involved in LysoPA acylation. I wonder if the expression levels and/or localization of AGPATs are altered by TANGO2 depletion.

2. Is the accumulation of reactive-oxygen species related to the lipid defects observed upon TANGO2 depletion? How? Do these cells have increased lipid peroxidation? This analysis may fall outside of the scope of this study but it would be important to discuss the potential links between the ROS and lipid changes.

3. Figure legends should be improved and include additional information. For example, in Figure 1A only a portion of the amino acid sequence is shown and the residue number should be included. Also, it is unclear what BAD/AVG/GOOD refers to.

*Reviewer #2 (Recommendations for the authors):*

This is an interesting study that adds significantly to our understanding of Tango2, a poorly characterized protein linked to muscular disease. Strengths include good use of cell biology and lipidomic analysis to profile tango2-depleted cells. A general concern is that while changes to lyso-lipids are observed, there is not yet a mechanistic understanding of these changes in tango2-depleted cells. Several intriguing models are proposed, but additional experiments are necessary to test these ideas. As noted below, there are also several sections that overstate their conclusions based on present data. Additional experiments should be conducted, or the conclusions toned down. Specific suggestions are below:

1. The authors note that the EGFP-tagged Tango2 localizes to both mitochondrial and the cytoplasm. Western blotting indicates the construct is not degraded, so they state "it is reasonable to conclude that the transmembrane-domain lacking TANGO2 is attached predominantly to the mitochondrial outer membrane…" This is an over-interpretation. The construct is being stable over-expressed, and the partial cytoplasmic localization may simply be due to overwhelming the mitochondrial import machinery. Further, investigate the topology of Tango 2 and/or tone down the conclusion.

2. RNAi-depletion of Tango2 causes LD accumulation, but it is unclear whether this is due to decreased LD lipolysis, increased LD biogenesis, or some combination of both. A possibility is the LD accumulation is due to enhanced conversion of PA to DAG, which can be used to make TAG in LDs. Lipidomics suggests a decrease in PA and accumulation of its precursor LPA, as well as changes/decreases in several other mature di-acyl lipids like ceramide. The overall model presented is that LPA and LPG are shunted into cardiolipin synthesis for mitochondrial, resulting in reduced steady-state PA levels. This is an intriguing model, but the data presented are not sufficient to demonstrate it. Additional radio-label or similar approaches to track lipogenesis and PA synthesis/turnover would substantially enhance this study. Since the study suggests a defect in fatty acyl-CoA metabolism, pulse-chase 14C-palmitate experiments would substantially help the work. 14C-palmitate tracking would also help determine if rates of acyl-CoA incorporation/turnover into phospholipids are defective in tango2-RNAi cells.

3. In overall model suggests lyso-lipids and enriched due to a defect in their acylation. This is done by LPAAT enzymes. Some work to examine whether Tango2 loss impacts LPAAT enzyme levels and/or transcription would help this study.

4. Similar to the above, the reduced PA levels in Tango2-depleted cells could be due to enhanced Lipin-mediated PA turnover into DAG. This DAG pool would then make TAG and more LDs, as observed. Examining Lipin expression/activity in Tango2-depleted cells would test this. Likewise, Lipin depletion may actually rescue some of the pathology of Tango2-depleted cells.

---

## [Author Response]

Essential revisions:1. If possible, please analyze the localization of endogenous TANGO2 by immunofluorescence using the antibody described in Figure S2.

We did not use commercial antibodies to analyze the endogenous localization of TANGO2 because, to our knowledge, no commercial antibodies for immunofluorescence (IF) have been developed. Please see the following table for the data on the commercial preparations used for detecting TANGO2. IHC stands for immunohistochemistry.

**Author response table 1. sa2table1:** 

Company	Antibody	Host	Clonality	Immunogen sequence	Applications
Merck	Anti-TANGO2 antibody	Rabbit	Polyclonal	TPVNVQRREDSATEGSHRLILAANRDEFYSRPSKLADFWGNNNEILSGLDMEEGKEGGTWLGISTRGKLAALTNYLQPQLDWQARGRGELVTHFLTTDVDSLSYLKKVSMEGHLYNGFNLIAADLSTAKGD VICYYGNRGEPDPIVLT	IHC
Invitrogen	TANGO2PolyclonalAntibody	Rabbit	Polyclonal	TPVNVQRREDSATEGSHRLILAANRDEFYSRPSKLADFWGNNNEILSGLDMEEGKEGGTWLGISTRGKLAALTNYLQPQLDWQARGRGELVTHFLTTDVDSLSYLKKVSMEGHLYNGFNLIAADLSTAKGDVICYYGNRGEPDPIVLT	IHC
Invitrogen	TANGO2PolyclonalAntibody	Rabbit	Polyclonal	Synthetic peptide directed towards the middle region ofhuman TANGO2	WB
Bioss	C22orf25PolyclonalAntibody	Rabbit	Polyclonal	KLH conjugated synthetic peptide derived from human C22orf25, amino acids 25-75.	IHC, WB
OriGene	C22orf25(TANGO2) PolyclonalAntibody	Rabbit	Polyclonal	MCIIFFKFDPRPVSKNAYRLILAANRDEFYSRPSKLADFWGNNNEILSGL	WB
NovusBiologicals	TANGO2Antibody	Rabbit	Polyclonal	MCIIFFKFDPRPVSKNAYRLILAANRDEFYSRPSKLADFWGNNNEILSGL	WB

The commercial antibody we use to determine the presence of TANGO2 by western blot is produced by Novus Biologicals.

The idea that TANGO2 localizes to membrane contact sites between mitochondria and the ER and LDs would also be strengthened by experiments including multiple organelle markers and/or methods to enhance the detection of membrane contact sites [https://doi.org/10.1073/pnas.1910854117].

We have addressed these concerns by the following two independent methods

1. FRET-based analysis of TANGO2 with Tom20 protein of the mitochondria (Figure 2)

2. Imaging TANGO2 location in fixed cells with specific ER and LDs proteins in all confocal planes (XY, XZ, and YZ) (Figure 2 and Figure 2—figure supplement 1).

2. Please display the absolute levels of all lipids in the various conditions.

Done (Figure 4—figure supplement 1).

3. RNAi-depletion of Tango2 causes LD accumulation, but it is unclear whether this is due to decreased LD lipolysis, increased LD biogenesis, or some combination of both. Pulse-chase 14C-palmitate experiments (or heavy label palmitate with mass spec) would substantially help the work and would also help determine if rates of acyl-CoA incorporation/turnover into phospholipids are defective in tango2-RNAi cells.

We performed many experiments with 14Cpalmitate pulse-chase t, but the assays were not sensitive to obtain any meaningful data. We then analyzed LD biogenesis and lipolysis using diethylumbelliferyl phosphate (DEUP) in control and starved conditions by flow cytometry (as described in doi:10.1016/j.cub.2013.06.032). We now explain this in the paper. As shown in Figure 3 (new panels H and I), TANGO2-depleted fibroblasts from TDD donors are altered in LD lipolysis and biogenesis.

4. Check whether Tango2 loss impacts AGPAT, LPAAT, and Lipin enzyme levels and/or transcription.

As both reviewers suggested, we measured mRNA levels of all GPAT, AGPAT, and LIPIN isoforms in Mock and TANGO2-depleted cells in control and starvation medium by RT-qPCR.The data are included in the paper.

Reviewer #1 (Recommendations for the authors):Below are additional specific comments/suggestions that may help improve the manuscript.1. The accumulation of lysoPA in Tango2-depleted cells may suggest a defect in AGPAT enzymes, involved in LysoPA acylation. I wonder if the expression levels and/or localization of AGPATs are altered by TANGO2 depletion.

We monitored the mRNA levels of AGPAT1-5 in Mock and TANGO2-depleted cells in the control and starvation medium by RT-qPCR. As shown in Figure 5, the mRNA levels of AGPAT1, AGPT2, AGPT3, and AGPT5 are altered in TANGO2-depleted cells. We have explained the significance of these changes in the paper. This is likely a feedback mechanism to maintain the levels of PA. But in cells lacking TANGO2, this signaling method is lost, which further adds to the overall problem of lipid homeostasis.

2. Is the accumulation of reactive-oxygen species related to the lipid defects observed upon TANGO2 depletion? How? Do these cells have increased lipid peroxidation? This analysis may fall outside of the scope of this study but it would be important to discuss the potential links between the ROS and lipid changes.

Good suggestion. We measured total lipid peroxidation in TANGO2-depleted HepG2 cells and human fibroblasts from TDD patients. As shown in the new panels (F-H) of Figure 6, lipid peroxidation increases in cells lacking TANGO2 compared to control cells. These results are now explained in the paper.

3. Figure legends should be improved and include additional information. For example, in Figure 1A only a portion of the amino acid sequence is shown and the residue number should be included. Also, it is unclear what BAD/AVG/GOOD refers to.

Done.

Reviewer #2 (Recommendations for the authors):This is an interesting study that adds significantly to our understanding of Tango2, a poorly characterized protein linked to muscular disease. Strengths include good use of cell biology and lipidomic analysis to profile tango2-depleted cells. A general concern is that while changes to lyso-lipids are observed, there is not yet a mechanistic understanding of these changes in tango2-depleted cells. Several intriguing models are proposed, but additional experiments are necessary to test these ideas. As noted below, there are also several sections that overstate their conclusions based on present data. Additional experiments should be conducted, or the conclusions toned down. Specific suggestions are below:

We thank you for your support and suggestions.

1. The authors note that the EGFP-tagged Tango2 localizes to both mitochondrial and the cytoplasm. Western blotting indicates the construct is not degraded, so they state "it is reasonable to conclude that the transmembrane-domain lacking TANGO2 is attached predominantly to the mitochondrial outer membrane…" This is an over-interpretation. The construct is being stable over-expressed, and the partial cytoplasmic localization may simply be due to overwhelming the mitochondrial import machinery. Further, investigate the topology of Tango 2 and/or tone down the conclusion.

We now include data on the topology of TANGO2 by Förster resonance energy transfer (FRET) microscopy. Based on our data, we can confidently state that TANGO2 and Tom20 of the mitochondrial membrane ranges are within 1 -10 nm of each other (Figure 2 panels B-D). We still detect a cytoplasmic pool and how this protein is both at the mitochondrial outer membrane and the cytoplasm remains untested.

2. RNAi-depletion of Tango2 causes LD accumulation, but it is unclear whether this is due to decreased LD lipolysis, increased LD biogenesis, or some combination of both. A possibility is the LD accumulation is due to enhanced conversion of PA to DAG, which can be used to make TAG in LDs. Lipidomics suggests a decrease in PA and accumulation of its precursor LPA, as well as changes/decreases in several other mature di-acyl lipids like ceramide. The overall model presented is that LPA and LPG are shunted into cardiolipin synthesis for mitochondrial, resulting in reduced steady-state PA levels. This is an intriguing model, but the data presented are not sufficient to demonstrate it. Additional radio-label or similar approaches to track lipogenesis and PA synthesis/turnover would substantially enhance this study. Since the study suggests a defect in fatty acyl-CoA metabolism, pulse-chase 14C-palmitate experiments would substantially help the work. 14C-palmitate tracking would also help determine if rates of acyl-CoA incorporation/turnover into phospholipids are defective in tango2-RNAi cells.

We have analyzed the LD biogenesis and lipolysis using diethylumbelliferyl phosphate (DEUP) in control and starved conditions by flow cytometry (as previously described in doi:10.1016/j.cub.2013.06.032). As shown in Figure 3 (new panels H and I), TANGO2-depleted fibroblasts from TDD donors are defective in both LD lipolysis and biogenesis.

3. In overall model suggests lyso-lipids and enriched due to a defect in their acylation. This is done by LPAAT enzymes. Some work to examine whether Tango2 loss impacts LPAAT enzyme levels and/or transcription would help this study.

We measured the mRNA levels of all LPAAT/AGPAT isoforms (1-5) in Mock and TANGO2depleted cells cultured in control and starvation medium by RT-qPCR. As shown in Figure 5, the mRNA levels of AGPAT1, AGPT2, AGPT3, and AGPT5 are altered in TANGO2-depleted cells. The data are explained in the revised manuscript.

4. Similar to the above, the reduced PA levels in Tango2-depleted cells could be due to enhanced Lipin-mediated PA turnover into DAG. This DAG pool would then make TAG and more LDs, as observed. Examining Lipin expression/activity in Tango2-depleted cells would test this. Likewise, Lipin depletion may actually rescue some of the pathology of Tango2-depleted cells.

We measured the mRNA levels of LIPIN1-4 in Mock and TANGO2-depleted cells cultured in control and starvation medium by RT-qPCR. As shown in Figure 5, the mRNA levels of LIPIN1-4 didn't have significant alterations in TANGO2-depleted cells. The data are included and explained in the revised manuscript.